# Different translation dynamics of β- and γ-actin regulates cell migration

Pavan Vedula[1], Satoshi Kurosaka[2], Brittany MacTaggart[1], Qin Ni[3], Garegin Papoian[4], Yi Jiang[5], Dawei W Dong[1,6], Anna Kashina[1]*

[1]Department of Biomedical Sciences, School of Veterinary Medicine, University of Pennsylvania, Philadelphia, United States; [2]Institute of Advanced Technology, Kindai University, Kainan, Wakayama, Japan; [3]Department of Chemical and Biomolecular Engineering, University of Maryland, College Park, United States; [4]Department of Chemistry, University of Maryland, College Park, United States; [5]Department of Mathematics and Statistics, Georgia State University, Atlanta, United States; [6]Institute for Biomedical Informatics, Perelman School of Medicine, University of Pennsylvania, Philadelphia, United States

**Abstract** β- and γ-cytoplasmic actins are ubiquitously expressed in every cell type and are nearly identical at the amino acid level but play vastly different roles in vivo. Their essential roles in embryogenesis and mesenchymal cell migration critically depend on the nucleotide sequences of their genes, rather than their amino acid sequences; however, it is unclear which gene elements underlie this effect. Here we address the specific role of the coding sequence in β- and γ-cytoplasmic actins' intracellular functions, using stable polyclonal populations of immortalized mouse embryonic fibroblasts with exogenously expressed actin isoforms and their 'codon-switched' variants. When targeted to the cell periphery using β-actin 3'UTR; β-actin and γ-actin have differential effects on cell migration. These effects directly depend on the coding sequence. Single-molecule measurements of actin isoform translation, combined with fluorescence recovery after photobleaching, demonstrate a pronounced difference in β- and γ-actins' translation elongation rates in cells, leading to changes in their dynamics at focal adhesions, impairments in actin bundle formation, and reduced cell anchoring to the substrate during migration. Our results demonstrate that coding sequence-mediated differences in actin translation play a key role in cell migration.

**\*For correspondence:** akashina@upenn.edu

**Competing interests:** The authors declare that no competing interests exist.

## Introduction

Actin is one of the most essential and abundant eukaryotic proteins, highly conserved across the tree of life. Among the six mammalian actins, β- and γ-cytoplasmic actins are the only two that are ubiquitously expressed in every mammalian cell type and share the highest identity at the amino acid level, with only four conservative substitutions within their N-termini (*Vandekerckhove and Weber, 1978*). Despite this near-identity, the coding sequences for these two actin isoforms are different by approximately 13% due to synonymous substitutions (*Erba et al., 1986*). Our work has previously shown that this coding sequence difference can lead to differential arginylation of these two actins. Arginylated β-actin accumulates in vivo, while arginylated γ-actin is degraded (*Zhang et al., 2010*).

In mice, β- and γ-cytoplasmic actins play vastly different physiological roles. β-actin knockout leads to defects in embryogenesis and early embryonic lethality (*Bunnell et al., 2011*; *Shawlot et al., 1998*; *Shmerling et al., 2005*; *Strathdee et al., 2008*; *Tondeleir et al., 2013*; *Tondeleir et al., 2014*), while γ-cytoplasmic actin knockout mice survive until birth and have much milder overall phenotypic defects (*Belyantseva et al., 2009*; *Bunnell and Ervasti, 2010*). Our prior work has shown that β-actin's nucleotide sequence, rather than its amino acid sequence, underlies

**eLife digest** Most mammalian cells make both β- and γ-actin, two proteins which shape the cell's internal skeleton and its ability to migrate. The molecules share over 99% of their sequence, yet they play distinct roles. In fact, deleting the β-actin gene in mice causes death in the womb, while the animals can survive with comparatively milder issues without their γ-actin gene. How two similar proteins can have such different biological roles is a long-standing mystery.

A closer look could hold some clues: β- and γ-actin may contain the same blocks (or amino acids), but the genetic sequences that encode these proteins differ by about 13%. This is because different units of genetic information – known as synonymous codons – can encode the same amino acid. These 'silent substitutions' have no effect on the sequence of the proteins, yet a cell reads synonymous codons (and therefore produces proteins) at different speeds.

To find out the impact of silent substitutions, Vedula et al. swapped the codons for the two proteins, forcing mouse cells to produce β-actin using γ-actin codons, and vice versa. Cells with non-manipulated γ-actin and those with β-actin made using γ-actin codons could move much faster than cells with β-actin. This suggested that silent substitutions were indeed affecting the role of the protein.

Vedula et al. found that cells read γ-codons – and therefore made γ-actin – much more slowly than β-codons: this also affected how quickly the protein could be dispatched where it was needed in the cell. Slower production meant that bundles of γ-actin were shorter, which allowed cells to move faster by providing a weaker anchoring system. Overall, this work provides new links between silent substitutions and protein behavior, a relatively new research area which is likely to shed light on other protein families.

its essential role in embryogenesis (*Vedula et al., 2017*). Using CRISPR/Cas9, we edited the five nucleotides at the beginning of the β-actin coding sequence within the β-actin gene (*Actb*), causing it to encode γ-actin protein (*Actb$^{c-g/c-g}$*, β-coded γ-actin). Such *Actb$^{c-g/c-g}$* mice developed normally and showed no gross phenotypic defects, thus demonstrating that the intact β-actin gene, rather than protein, defines its essential role in embryogenesis (*Patrinostro et al., 2018*; *Vedula et al., 2017*). Thus, nucleotide sequence constitutes a major, previously unknown determinant of actin function, even though it is unclear which specific nucleotide-based elements of the actin gene play a role in this effect.

Here we tested whether the coding sequence alone, in the context of invariant non-coding elements and independent of the positional effects of the actin gene, plays a role in cytoplasmic actins' intracellular function. To do this, we incorporated the coding sequences of β- and γ-actins and their 'codon-switched' variants (β-coded γ-actin and γ-coded β-actin) into otherwise identical constructs containing the human β-actin promoter, an N-terminal enhanced Greef Fluorescent Protein (eGFP) fusion, and the β-actin 3′UTR. Stable expression of these constructs in mouse embryonic fibroblasts (MEFs) resulted in dramatically different effects on directional cell migration. While cells expressing β-actin migrated at rates similar to wild-type untransfected cells in wound-healing assays, cells expressing γ-actin migrated nearly twofold faster. This difference depended directly on the coding sequence, as evident by the use of the 'codon-switched' actin variants. Expression of γ- or γ-coded β-actin led to changes in cell morphology and distribution of focal adhesions, which were larger in size and localized mostly at the cell periphery rather than under the entire cell, similarly to previously reported cellular phenotypes linked to poorer cell attachment and faster migration (*Kim and Wirtz, 2013*). Focal adhesions in γ-actin- or γ-coded-β-actin-expressing cells appeared to be poorly anchored, though often not visibly associated with long actin bundles. In contrast, long actin cables could be clearly seen anchoring focal adhesions in β-actin- or β-coded γ-actin-expressing cells.

Single-molecule measurements of actin translation using the SunTag system at the focal adhesion sites showed an approximately twofold faster translation elongation of β-actin compared to γ-actin. Fluorescence recovery after photobleaching (FRAP) demonstrated that γ-actin accumulation in cells was slower than β-actin, further confirming global differences in actin isoform translation. Molecular simulations of actin assembly at the focal adhesions showed that differences in translation rates can

directly impact actin bundle formation, leading to shorter actin bundles in the case of slower translating γ-actin, in agreement with our experimental data.

Our results demonstrate that nucleotide coding sequence-dependent translation rates, coupled to zipcode-targeted *actin* mRNA localization, play an essential role in differentiating actin isoforms' function in cell migration.

## Results

### β- and γ-actin coding sequences have differential effects on cell migration speed

To test the specific effect of coding sequences on intracellular functions of actin isoforms, we generated immortalized MEF cell cultures stably expressing β- and γ-actin coding sequences, as well as their codon-switched variants (β-coded γ-actin and γ-coded β-actin), cloned into identical expression constructs under the human β-actin promoter, containing an N-terminal eGFP fusion and the β-actin 3'UTR (*Figure 1*, top left, and 'Materials and methods'). This construct design enabled us to confine our experiments to the effects of the coding sequence and exclude any potential contribution from other elements known to mediate differences between β- and γ-actins, including promoter-mediated transcription (*Tunnacliffe et al., 2018*), differential 3'UTR-mediated mRNA targeting (*Hill and Gunning, 1993*; *Katz et al., 2012*; *Kislauskis et al., 1993*; *Kislauskis et al., 1994*), and differential N-terminal processing (*Zhang et al., 2010*). Cell populations stably expressing eGFP constructs were checked to ensure similar levels of *eGFP* mRNA, as well as to confirm that the expression of the exogenous eGFP-actin did not have any significant effect on the endogenous *β-actin* and *γ-actin* mRNA levels (*Figure 1—figure supplement 1*). We also confirmed that β-actin 3'UTR targeted the *eGFP-actin* mRNA to the cell periphery, using fluorescence in situ hybridization (FISH) (*Figure 1—figure supplement 2*). Finally, we confirmed that the level and distribution of F-actin in each of the cell cultures transfected with different actin isoforms was largely similar to each other (*Figure 1—figure supplement 3*). Thus, in these cell populations, the effects of exogenously expressed actin could be tested without perturbation of other actin-related processes that are essential for cell viability.

β- and γ-cytoplasmic actins make up more than 50% of the total actin in these cells and have been previously shown to play major non-overlapping roles in directional cell migration (*Patrinostro et al., 2017*). We, therefore, tested whether cells expressing eGFP-β-actin or eGFP-γ-actin showed any differences in cell migration using a wound-healing assay. Strikingly, while cells expressing eGFP-β-actin migrated at rates similar to wild-type untransfected cells, cells expressing eGFP-γ-actin migrated nearly twofold faster (*Figure 1*, top right and bottom; *Figure 1—figure supplement 4*; *Videos 1* and *2*). This difference in cell migration rates was coding sequence dependent, as seen in cells expressing the codon-switched actin variants, γ-coded β-actin (which migrated faster, similarly to those expressing γ-actin), and β-coded γ-actin (which migrated slower, like β-actin-expressing cells) (*Figure 1*, top left; *Videos 3* and *4*). Thus, the effect of actin isoform expression on directional cell migration is mediated by their nucleotide coding sequence and does not appear to be influenced by their amino acid sequence.

In normal cells, the *β-actin* 3'UTR contains a zipcode sequence that is required for its mRNA localization to the cell periphery (*Kislauskis et al., 1993*) and has been shown to be important for directional cell migration (*Condeelis and Singer, 2005*; *Katz et al., 2012*; *Kislauskis et al., 1994*; *Kislauskis et al., 1997*). *γ-actin* mRNA has no such sequence and does not undergo targeting to the cell periphery (*Hill and Gunning, 1993*). All our constructs described above contained the β-actin 3'UTR with the zipcode sequence as one of the constant elements. To test whether 3'UTR-mediated targeting of *actin* mRNA affects the cell migration phenotypes observed in our stably transfected cell cultures, we performed the same experiment using cell cultures stably expressing similar actin constructs, but without the β-actin 3'UTR (*Figure 1—figure supplement 2*). These cells did not exhibit significant differences in cell migration rates (*Figure 1—figure supplement 5*). Thus, differences in the effects of cytoplasmic actin coding sequences on cell migration require mRNA targeting to the cell periphery.

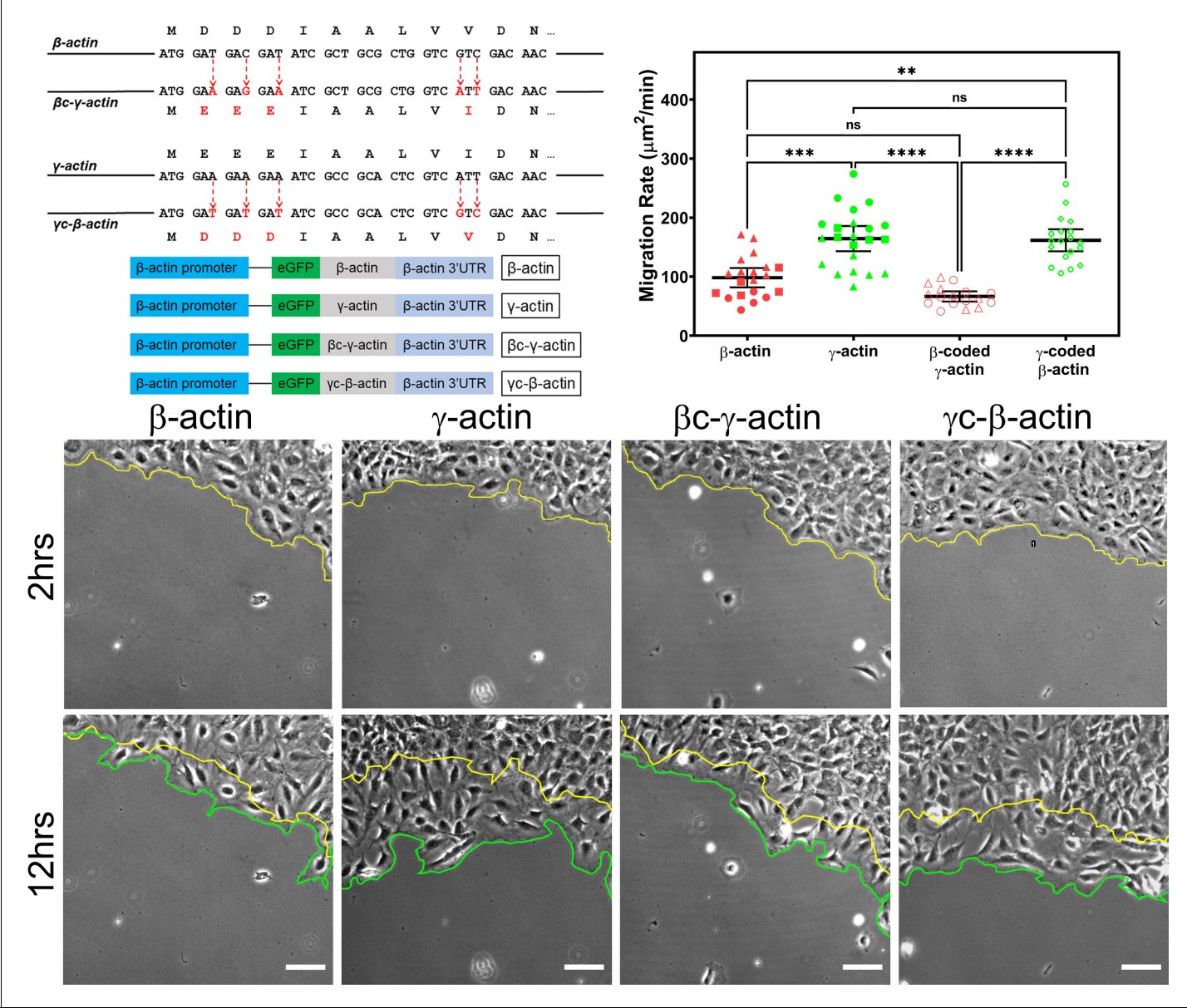

**Figure 1.** Cell migration speed is regulated by actin isoform coding sequence. Top left, mutagenesis strategy used to generate 'codon-switched' actin variants and linear maps of the major constructs used in this study. Top right, scatter plot of migration rates in a wound-healing assay of cell populations expressing different actin constructs. Bottom, representative images of the migrating wound edge, with the initial and the final position of the edge denoted by the yellow and the green line, respectively. Cell migration rates were derived as the area covered over time by the cell layer within the field of view (calculated as the area between the yellow and the green lines). N = 20 (for β-actin); 22 (for γ-actin); 18 (for βc-γ-actin); 19 (for γc-β-actin). Independent experimental replicates (two or more) are indicated by different symbols within each cell population. One way non-parametric analysis of variance (ANOVA) yielded a p-value<0.0001 with multiple comparisons shown on the plot. Error bars represent mean ± 95% CI. *p<0.05, **p<0.01, ***p<0.001, and ****p<0.0001. See also *Videos 1–4*. Scale bars, 100 μm.

The online version of this article includes the following figure supplement(s) for figure 1:

**Figure supplement 1.** Cell transfection with actin isoforms does not perturb the endogenous levels of actin mRNA.

**Figure supplement 2.** Transfected actin isoform mRNAs are targeted to the cell periphery via 3'UTR.

**Figure supplement 3.** Cell transfection with actin isoforms does not perturb the level or distribution of the actin polymer.

**Figure supplement 4.** Expressing γ-actin with β-actin 3'UTR increases cell migration rate compared to expressing β-actin with β-actin 3'UTR.

**Figure supplement 5.** Expressing γ-actin with β-actin 3'UTR increases cell migration rate.

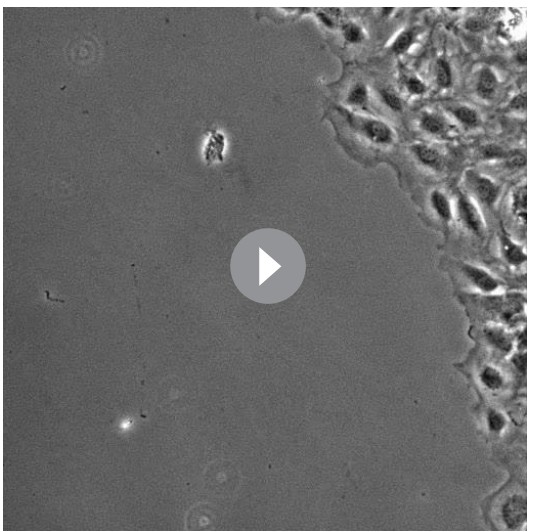

**Video 1.** Migration of β-actin-transfected cells. Mouse embryonic fibroblast (MEF) cell cultures stably expressing eGFP-β-actin migrating into an infinite wound edge over a period of 10 hr.
https://elifesciences.org/articles/68712#video1

# Actin isoforms affect focal adhesion size, cell spreading, and actin dynamics at focal adhesions in a coding sequence-dependent manner

Changes in cell migration rates are normally associated with changes in actin dynamics at the leading edge, rate and persistence of leading edge protrusions and retractions, as well as focal adhesion formation and dynamics, which affect cell spreading, polarization, and attachment to the substrate. Focal adhesions' strength and persistence are closely regulated by their association with actin filaments, which grow at the focal adhesion sites to form a dynamic actin bundle that participates in anchoring the cells to the substrate. Thus, focal adhesions critically depend on actin dynamics in the vicinity of the adhesion site. In turn, focal adhesions can regulate cell spreading and polarization, in addition to cell migration rates.

To test these processes in eGFP-actin isoform-transfected cells, we first looked at the rate and persistence of leading edge protrusions and retractions, but found no consistent differences between the cell populations that correlated with cell migration rates (*Figure 2—figure supplement 1*). We next assessed focal adhesion dynamics in these cells using total internal reflection fluorescence microscopy (TIRF-M) of eGFP-β-actin and eGFP-γ-actin. Since the imaging volume in TIRF-M is limited to the basal 300 nm or less, we reasoned that most of the actin signals visible in this volume should be associated with focal adhesion patches. Imaging the long-term (hours) behavior of actin at focal adhesion patches during wound healing using TIRF-M revealed that in migrating cells, eGFP-β-actin patches appeared more prominent and

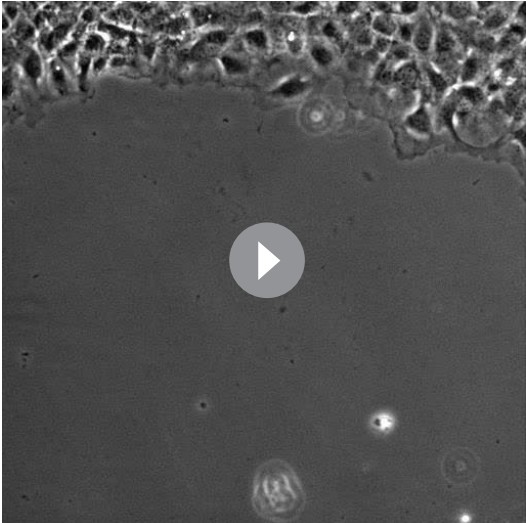

**Video 2.** Migration of γ-actin-transfected cells. Mouse embryonic fibroblast (MEF) cell cultures stably expressing eGFP-γ-actin migrating into an infinite wound edge over a period of 10 hr.
https://elifesciences.org/articles/68712#video2

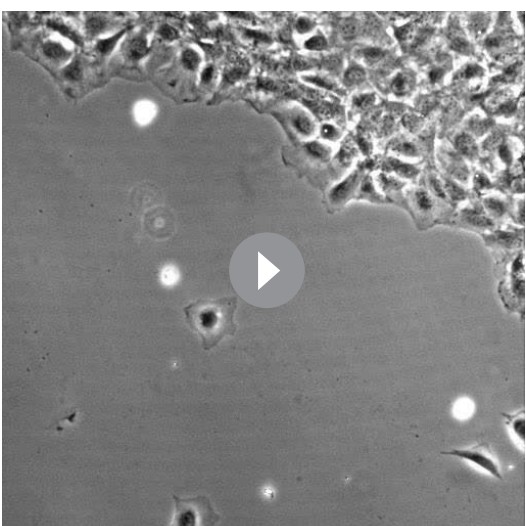

**Video 3.** Migration of β-coded-γ-actin-transfected cells. Mouse embryonic fibroblast (MEF) cell cultures stably expressing eGFP-β-coded γ-actin migrating into an infinite wound edge over a period of 10 hr.
https://elifesciences.org/articles/68712#video3

persisted considerably longer than eGFP-γ-actin patches (*Figure 2A–C*), suggesting that focal adhesions in eGFP-β-actin-expressing cells persist for longer periods of time. At the same time, testing short-term (5 min) actin dynamics at the focal adhesions using Fluorescence Recovery After Photobleaching (FRAP) showed no notable differences in focal adhesion recovery rates that correlated with either coding or amino acid sequence (*Figure 2—figure supplement 2*). Thus, different actin isoforms affect long-term focal adhesion persistence without strongly affecting short-term focal adhesion or protrusion dynamics during persistent directional migration at the cell leading edge.

To get deeper insights into the focal adhesion morphology and distribution in cells transfected with different actin isoforms, we grew cells at a low density, to enable visualization of the morphology and cytoskeleton-dependent structures in individual cells isolated on coverslips, without contacting their neighbors. Notably, cells in such scarce cultures are under no stimuli to migrate. Many of them remain stationary or move randomly around the same area, resulting in much slower rates of persistent migration and overall displacement over time. Consequently, such sparsely grown cells transfected with different actin isoforms do not prominently differ from each other in their migration (*Figure 3—figure supplement 1*), even though they are expected to undergo similar actin isoform-related changes at the subcellular level.

To analyze focal adhesions and spreading in these cells, we first used TIRF-M to image single cells stained with antibodies to the focal adhesion protein paxillin. These assays revealed prominent differences in focal adhesion morphology and distribution between the cell populations transfected with different actin isoforms (*Figure 3*, top row of images; see also *Figure 3—figure supplements 2–5*). In eGFP-β-actin-expressing cells, focal adhesions had a normal elongated morphology and were distributed throughout the entire cell footprint. In contrast, eGFP-γ-actin-expressing cells formed focal adhesions that localized mostly at the cell periphery (*Figure 3*, top row of images; *Figure 3—figure supplement 6A,B*). This trend depended on the actin coding sequence: focal adhesions in eGFP-β-coded-γ-actin-expressing cells resembled those in eGFP-β-actin, while focal adhesions in eGFP-γ-coded-β-actin-expressing cells were like those in eGFP-γ-actin-expressing cells (*Figure 3—figure supplement 6A,B*).

Imaging eGFP-actin in widefield showed that most focal adhesions in cells expressing eGFP-β-actin and eGFP-β-coded γ-actin were associated with long thick bundles of actin emanating from the focal adhesion point (*Figure 3*, top, and *Figure 3—figure supplements 2–5*). In comparison, the dorsal bundles connecting to the focal adhesions were much less prominent in γ-actin- and γ-coded β-actin-expressing cells.

Focal adhesion size uniquely predicts cell migration rate (*Kim and Wirtz, 2013*), with the larger focal adhesions correlating with faster migration speeds. We measured the focal adhesion area in all the four cell cultures transfected with different actin isoforms and found that faster migrating cells expressing γ-actin, and γ-coded β-actin indeed, had significantly larger focal adhesions than slower migrating cells expressing β-actin and β-coded γ-actin (*Figure 3*, bottom). Morphologically, focal adhesions in γ-actin- and γ-coded β-actin-expressing cells appeared wider and less elongated than in β-actin- and β-coded γ-actin-expressing cells; however, global measurements of their aspect ratios did not reveal any consistent statistically significant differences (*Figure 3—figure supplement 6C*). This could be due to the fact that the majority of focal adhesions in all of these cells were small and dot-like, and only a few larger ones tended to exhibit potential differences in morphology. Thus, β-

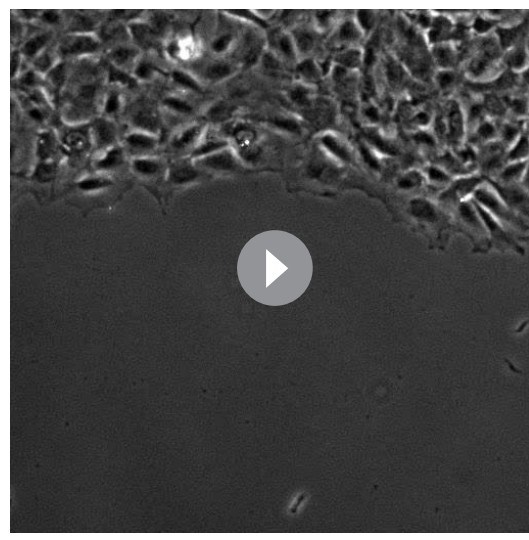

**Video 4.** Migration of γ-coded-β-actin-transfected cells. Mouse embryonic fibroblast (MEF) cell cultures stably expressing eGFP-γ-coded β-actin migrating into an infinite wound edge over a period of 10 hr. https://elifesciences.org/articles/68712#video4

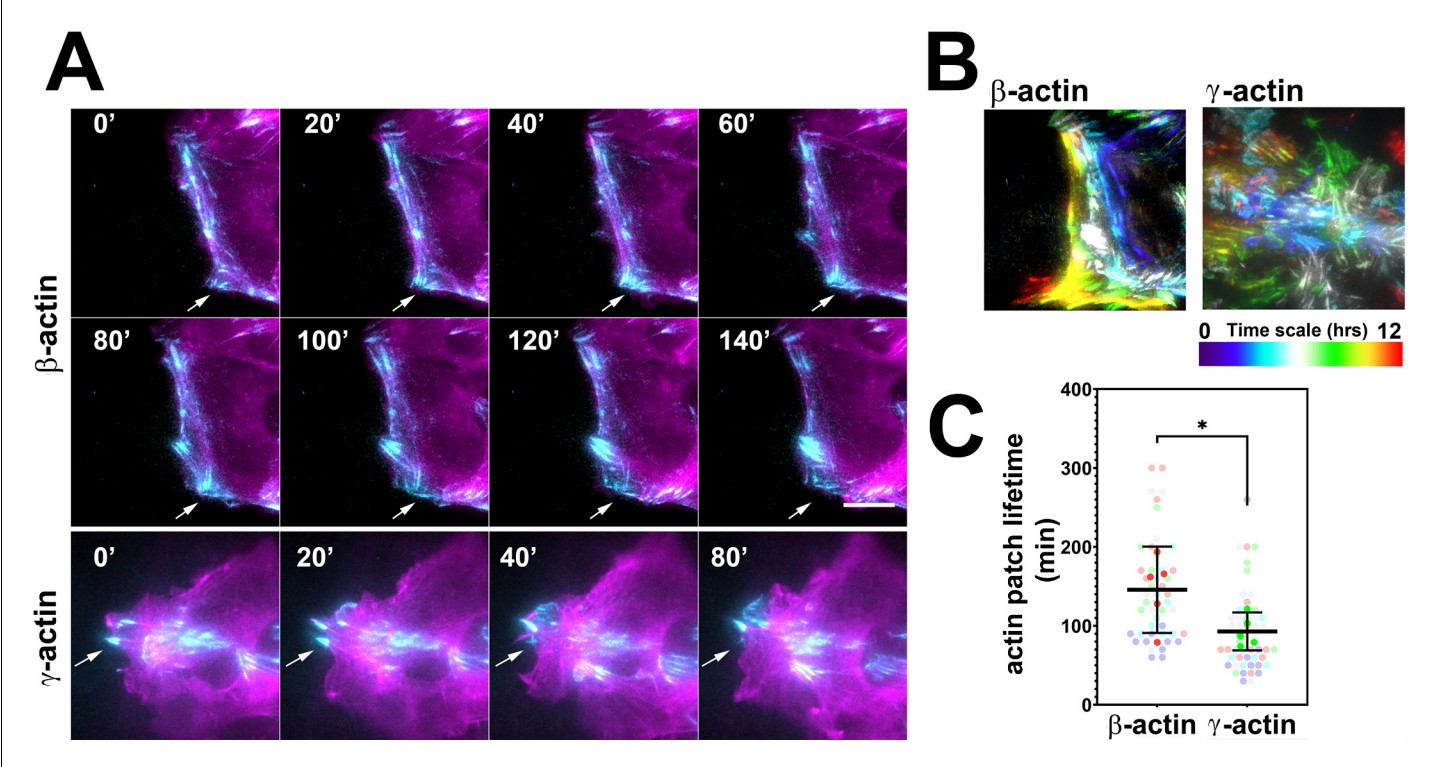

**Figure 2.** Actin isoforms have differential effects on focal adhesion dynamics. (**A**) Montages of eGFP-β-actin- (top two rows) and eGFP-γ-actin (bottom row)-expressing cells undergoing wound healing following a scratch wound. Total internal reflection fluorescence microscopy (TIRF-M) images (cyan) are overlaid with widefield images (magenta). Scale bar = 10 µm. Arrows point to focal adhesions being formed and disassembled over time. (**B**) Maximum intensity projections of TIRF-M images of eGFP-actin over time for each of the β- and γ-actin-expressing cell populations during a 12-hr wound healing. Scale represents the temporal color scale. Note that β-actin persists longer in the TIRF plane as compared to γ-actin. (**C**) Actin patch lifetimes in the TIRF-M channel. Five fields of view for each cell population were used to estimate lifetimes (solid colors) with N = 50 (β-actin) and 53 (γ-actin) patches (transparent colors). Error bars represent mean ± 95% CI. Unpaired t-test gave a p-value<0.05.
The online version of this article includes the following figure supplement(s) for figure 2:

**Figure supplement 1.** Protrusion and retraction dynamics of cells at wound edge during wound healing.

**Figure supplement 2.** FRAP recovery curves for actin patches in cells at wound edge during wound healing imaged in TIRF-M.

and γ-actin coding sequences determine the size and distribution of focal adhesions in migrating cells in a manner that correlates with changes in their migration speed.

We also measured focal adhesion recovery rates in single-cell cultures using FRAP. Focal adhesions in cells transfected with β-actin and β-coded γ-actin recovered slightly faster than cells transfected with γ-actin and γ-coded β-actin (*Figure 3—figure supplement 7*). While statistically significant, these coding sequence-dependent differences were small, and thus it is unclear if they can prominently contribute to the cells' phenotype.

Cell spreading and polarization are critically determined by their adhesion to the substrate and correlate with their migratory behavior (*Kim and Wirtz, 2013*). To test whether focal adhesion changes in our cell populations are accompanied by changes in cell spreading and polarization, we used Celltool (*Pincus and Theriot, 2007*) to analyze shape distribution of single cells stably expressing eGFP-β-actin, eGFP-γ-actin, and their codon-switched variants. Using images of live single cells, the shape space was parameterized into various shape modes, and shape modes 1 and 2 accounted for ~60% of variance in shapes across all cells (*Figure 3—figure supplement 8A*, inset). The first mode roughly captures the variation in the size of the cell footprint on the substrate and accounts for ~40% of the variance in shape, while the second mode captures cell polarization and accounts for ~20% of the variance in shape. Using these shape modes to analyze images of cells expressing different actin isoforms' coding sequences, we found that cells expressing eGFP-β-actin had a larger footprint (*Figure 3—figure supplement 8A*, clustered to the left of the y-axis) and had more

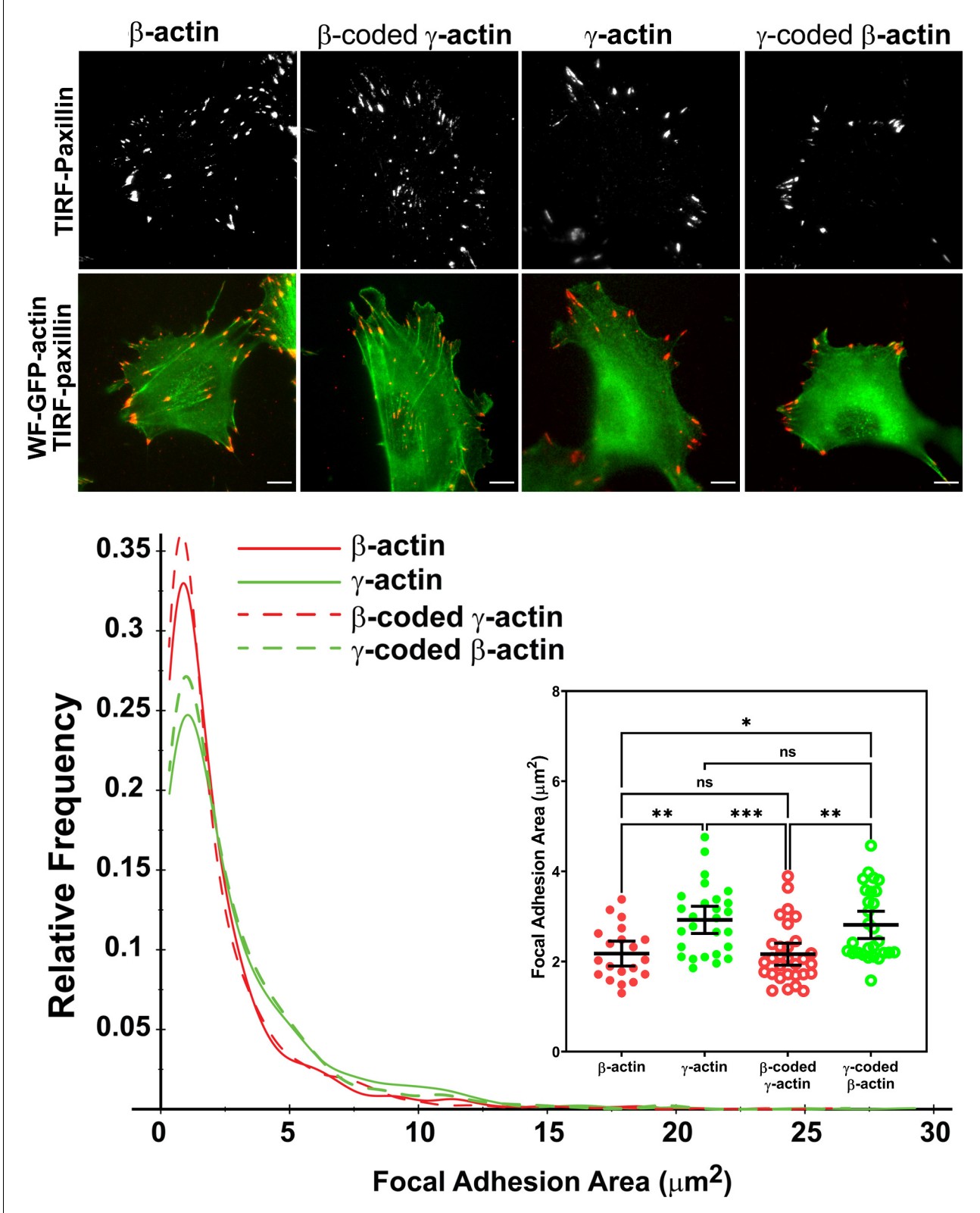

**Figure 3.** Actin isoform coding sequence affects the overall size and distribution of focal adhesions. Top, representative images of eGFP-actin-transfected cells stained with anti-paxillin to visualize focal adhesions. Total interference reflection fluorescence microscopy (TIRF-M) images of paxillin staining are shown alone (top) and as an overlay with widefield eGFP signals (bottom). Bottom, quantification of focal adhesion size in the four different cell populations, shown as a distribution plot (for all focal adhesions analyzed) and as a scatter plot (for individual cells) in the inset. Scale bar = 10 μm.

*Figure 3 continued on next page*

*Figure 3 continued*

β-actin = 20, γ-actin = 28, β-coded γ-actin = 30, and γ-coded β-actin = 29 cells with N = 996, 786, 1245, and 1041 focal adhesions for the respective cell types. Results for non-parametric one-way analysis of variance (ANOVA) with multiple comparisons are indicated on the graph.

The online version of this article includes the following figure supplement(s) for figure 3:

**Figure supplement 1.** Single-cell motility is not dependent on actin isoform coding or amino acid sequences.

**Figure supplement 2.** Representative images of paxillin and eGFP-actin distribution in cells stably transfected with eGFP-β-actin.

**Figure supplement 3.** Representative images of paxillin and eGFP-actin distribution in cells stably transfected with eGFP-γ-actin.

**Figure supplement 4.** Representative images of paxillin and eGFP-actin distribution in cells stably transfected with eGFP-γ-coded β-actin.

**Figure supplement 5.** Representative images of paxillin and eGFP-actin distribution in cells stably transfected with eGFP-β-coded γ-actin.

**Figure supplement 6.** Quantification of focal adhesion distribution and shape.

**Figure supplement 7.** FRAP recovery curves for actin patches in single-cell cultures imaged in TIRF-M.

**Figure supplement 8.** Actin isoforms confer differential effects on cell shape and spreading.

---

variance in their polarization (*Figure 3—figure supplement 8A*, spread across the y-axis), while cells expressing eGFP-γ-actin exhibited the opposite trends (*Figure 3—figure supplement 8A*, clustered mostly in the top right quadrant). Expression of the codon-switched actin variants, β-coded γ-actin, and γ-coded β-actin showed that the footprint size depended on the actin isoform coding sequence, while the polarization variance appeared to be amino acid sequence-dependent.

Changes in the area of the cell footprint can arise due to either reduced cell spreading or reduced overall cell size. To distinguish between these possibilities, we quantified the area of trypsinized near-spherical cells (pre-spreading), which directly reflects cell size and volume. Cells expressing the γ-actin coding sequence were slightly smaller than those expressing β-actin coding sequence (*Figure 3—figure supplement 8B*, left). This ~6% difference in cell size was far less prominent than the difference in spread cell area (*Figure 3—figure supplement 8B*, right), which accounted for a greater than 80% change in the size of cell footprint. Thus, cells expressing γ-actin are less spread on the substrate, and this difference in spreading is coding sequence-dependent.

## β-actin exhibits faster intracellular translation elongation than γ-actin

In search of an underlying mechanism that could link actin isoforms' coding sequence to their intracellular properties, we turned to our previous study that used computational predictions of the mRNA secondary structures for *β-actin* and *γ-actin*. This study suggested that the coding region of *β-actin* mRNA forms a more relaxed secondary structure than that of *γ-actin*, predicting potential differences in translation elongation rates (*Zhang et al., 2010*). Such differences, if prominent enough, could in principle lead to changes in cells' ability to form focal adhesions and migrate. To test this prediction, we first compared the rates of overall protein accumulation of eGFP-β- and eGFP-γ-actin, by comparing FRAP of the total eGFP signal in the cell after whole-cell photobleaching. We reasoned that this would serve as a proxy for estimation of newly synthesized β- and γ-actin (*Figure 4A*). Notably, the recovery observed in these FRAP experiments within a 10-min imaging window arises from the folding and maturation of already synthesized eGFP fused to actin (since the eGFP maturation rate in vivo has been estimated to be approximately 14 min [*Balleza et al., 2018*; *Iizuka et al., 2011*]); given the constant time delay, this recovery rate directly reflects the rate of de novo synthesized actin accumulation within the imaging window. Photobleaching was calibrated to ensure that the cells remained healthy and visually normal during the experiment (*Figure 4A*, bottom). The recovery rate was significantly faster for eGFP-β-actin compared to eGFP-γ-actin (*Figure 4A*, top right). Thus, newly synthesized β-actin accumulates in cells faster than γ-actin.

To directly estimate β- and γ-actin translation elongation rates, we performed single-molecule imaging of nascent peptide synthesis (SINAPS) for these two actin isoforms using the SunTag system (*Wu et al., 2016*). Similarly to the constructs used for generating eGFP-actin stable cell populations, we ensured that the coding sequence was the only variable, flanked by otherwise identical upstream and downstream elements, including the promoter of the polyubiquitin gene (UbC) for constitutive expression, the N-terminal 5′ SunTag fusion to visualize the nascent peptide, the C-terminal auxin-induced degron to degrade fully synthesized polypeptides and reduce the background signal, the β-actin 3′UTR for cell periphery targeting, and MS2 repeats in the non-coding region to visualize mRNA via constitutively expressed MS2 coat-binding protein (MCP) fused to a HaloTag (*Figure 4—figure supplement 1*).

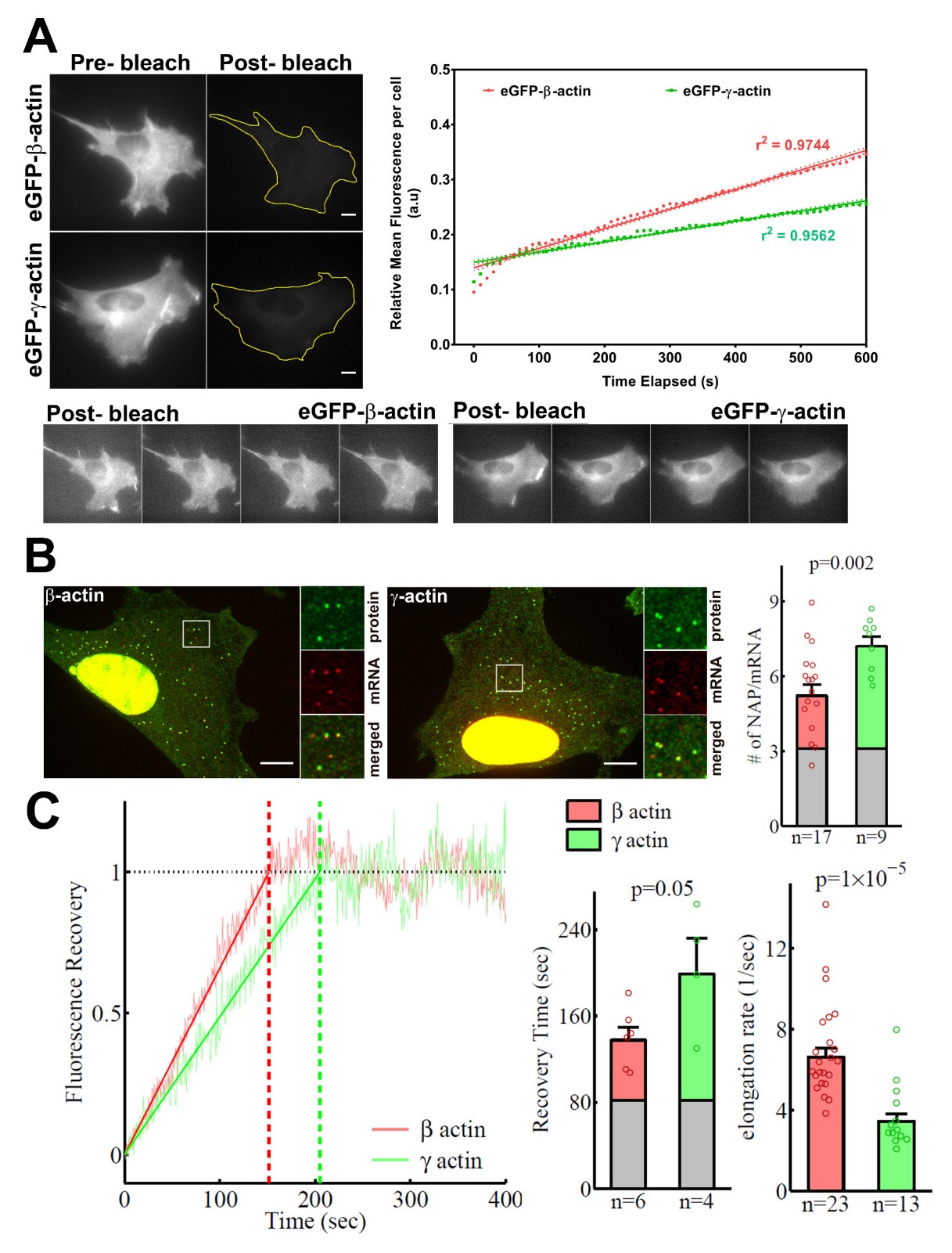

**Figure 4.** β-actin accumulates in cells faster than γ-actin and exhibits faster translation elongation. (**A**) Images and quantification of GFP fluorescence in cells before and after photobleaching the entire intracellular GFP signal. Top left panels show representative cells before and after photobleaching, with gray levels scaled to show the actual difference in signal intensity. Top right graph shows the fluorescence recovery after photobleaching (FRAP) curve during the first 10 min post-bleach. Data points are dots and the linear regression curves are in bold (with dotted lines representing the 95%

*Figure 4 continued on next page*

*Figure 4 continued*

confidence bands). N = 8 for eGFP-β-actin and N = 10 for eGFP-γ-actin. Bottom, post-bleached images of cells taken at 2.5′ intervals from 0′ (left) to 10′ (right), with the gray levels scaled up to enhance the residual eGFP signal. Scale bars = 10 μm. (B) Left: representative images of actin protein and mRNA used for the experiments. Insets on the right of each image show the enlarged region indicated in the image on the left. Right: quantification of ribosomes per mRNA, calculated as the SunTag signal from the nascent peptides (NAPs) at each translation site divided by the HaloTag signal from the mRNA. (C) Left: FRAP curves showing recovery of the green fluorescence signal in individual translation spots. Right: fluorescence recovery time for each actin isoform and translation elongation rate (amino acids/s) calculated from the fluorescence recovery and NAPs/RNA. (B) and (C) right: gray boxed area at the bottom of each bar indicates the contribution from SunTag, the auxin-induced degron (AID), and linker portion of the construct (see 'Materials and methods'), and p-values are from one-tailed Welch's t-test. Error bars represent SEM for n data points, calculated geometrically and plotted in a linear fashion.

The online version of this article includes the following figure supplement(s) for figure 4:

**Figure supplement 1.** Illustration of single-molecule imaging of nascent peptide synthesis for β- and γ-actin.

**Figure supplement 2.** Frequency distribution of the number of nascent peptides per mRNA on SINAPS constructs of β-actin (red) or γ-actin (green).

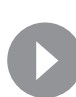

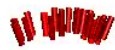

**Video 5.** Simulation of actin bundle growth from a focal adhesion at a slower subunit supply rate (40 subunits/s).

https://elifesciences.org/articles/68712#video5

Simultaneous imaging of SunTag-bound superfolder GFP (sfGFP) and MS2-bound Halo-Tag in fixed cells (labeled with JaneliaFluor 646) enabled us to estimate the number of nascent peptides (NAPs) per mRNA, a measure of the ribosome load and, by proxy, the translation elongation rate. Assuming similar translation initiation rates for both constructs (given their identical 5′ sequences at and around the translation initiation sites), fewer elongating ribosomes in this assay arise from their faster translocation over the mRNA, leading to weaker sfGFP signals per mRNA. Thus, differences in ribosome load per mRNA (and thus, the number of NAPs per mRNA) would directly indicate differences in translation elongation.

*γ-actin* coding sequence showed a nearly twofold higher level of NAPs/mRNA than that of *β-actin* (*Figure 4B*, *Figure 4—figure supplement 2*), indicating that the elongating ribosomes had a twofold higher load, and thus slower translocation rate, over *γ-actin* mRNA compared to *β-actin*. This is the first characterization of different ribosome loads on individual mRNAs of two protein isoforms that have the same coding sequence length.

Next, we estimated the real-time translation elongation rate of β- and γ-actin using FRAP of individual translation sites. To minimize mRNA movement, we tethered *SINAPS-β-actin* and *SINAPS-γ-actin* mRNA to focal adhesions, by co-transfecting cells with vinculin-MCP-HaloTag fusion. Fluorescence recovery rate of individual translation sites directly reflects the rate of translation elongation to generate new nascent peptides bearing new sfGFP bound to the SunTag peptides. This recovery rate was approximately twofold slower for γ-actin

compared to β-actin (*Figure 4C*), confirming that γ-actin translation elongation is indeed slower than that of β-actin, in agreement with the NAP/mRNA measurements.

Using both sets of data, we conclude that the translation elongation rate of the two actin isoforms differs by approximately twofold—faster for β-actin compared to γ-actin (*Figure 4D*).

## Faster initial subunit supply rates at the focal adhesions facilitate longer actin bundle formation

During cell migration, the initial formation of nascent focal adhesions critically depends on local actin subunit supply rate. Many studies assume that this subunit supply rate is not a limiting factor in vivo, due to high concentrations of G-actin at the cell leading edge (*Raz-Ben Aroush et al., 2017*). However, it is likely that the actin bundles forming at focal adhesion sites must compete with the actin meshwork at the lamellipodium for polymerization competent actin. There is increasing evidence suggesting such a competition between various actin-driven processes in vivo (*Faust et al., 2019*; *Suarez and Kovar, 2016*). In addition, it is possible that at a given moment, some, or most, of the free actin can be sequestered, for example, by monomer-binding proteins (*Skruber et al., 2018*),

forcing the elongating leading edge filaments to depend on de novo synthesized actin. In support, *actin* mRNA targeting to the cell leading edge is essential for cell migration, suggesting that local actin synthesis at the cell leading edge must be important (*Katz et al., 2012*). Furthermore, local actin translation bursts have been observed in neurons (*Buxbaum et al., 2014*). It is possible that these bursts, regardless of the overall actin concentrations, are required for locally supplying actin subunits at the focal adhesions during cell migration. If so, replacing the faster translationally elongating β-actin with the slower elongating γ-actin at these sites could potentially limit this supply and make a difference in focal adhesion anchoring, leading to shorter actin bundles at the focal adhesions, poorer spreading, and faster migration seen in γ-actin-expressing cells.

Since measuring local polymerization-competent actin in a cell is impossible experimentally, we used the computational model of active networks, MEDYAN (*Popov et al., 2016*). We simulated actin bundle growth at the focal adhesions at different subunit supply rates, in the presence of non-muscle myosin II motors and α-actinin as crosslinkers, which are critical for actin filament bundling in cells (*Chandrasekaran et al., 2019*; *Figure 5A* and *Figure 5—figure supplement 1A*). In these simulations, filaments elongate by incorporating newly supplied actin monomers, and then bundle together through the action of myosin motors and crosslinkers. During 10-min simulations, the time window typically sufficient for establishment of robust focal adhesions, varying actin subunit supply rate resulted in pronounced differences in the length of the actin bundle growing from the focal adhesion site (*Videos 5* and *6*). A twofold decrease in subunit supply rate resulted in over a twofold decrease in actin bundle length (*Figure 5A*, right and *Figure 5—figure supplement 1B*). To test this

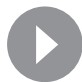

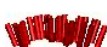

**Video 6.** Simulation of actin bundle growth from a focal adhesion at a faster subunit supply rate (60 subunits/s).
https://elifesciences.org/articles/68712#video6

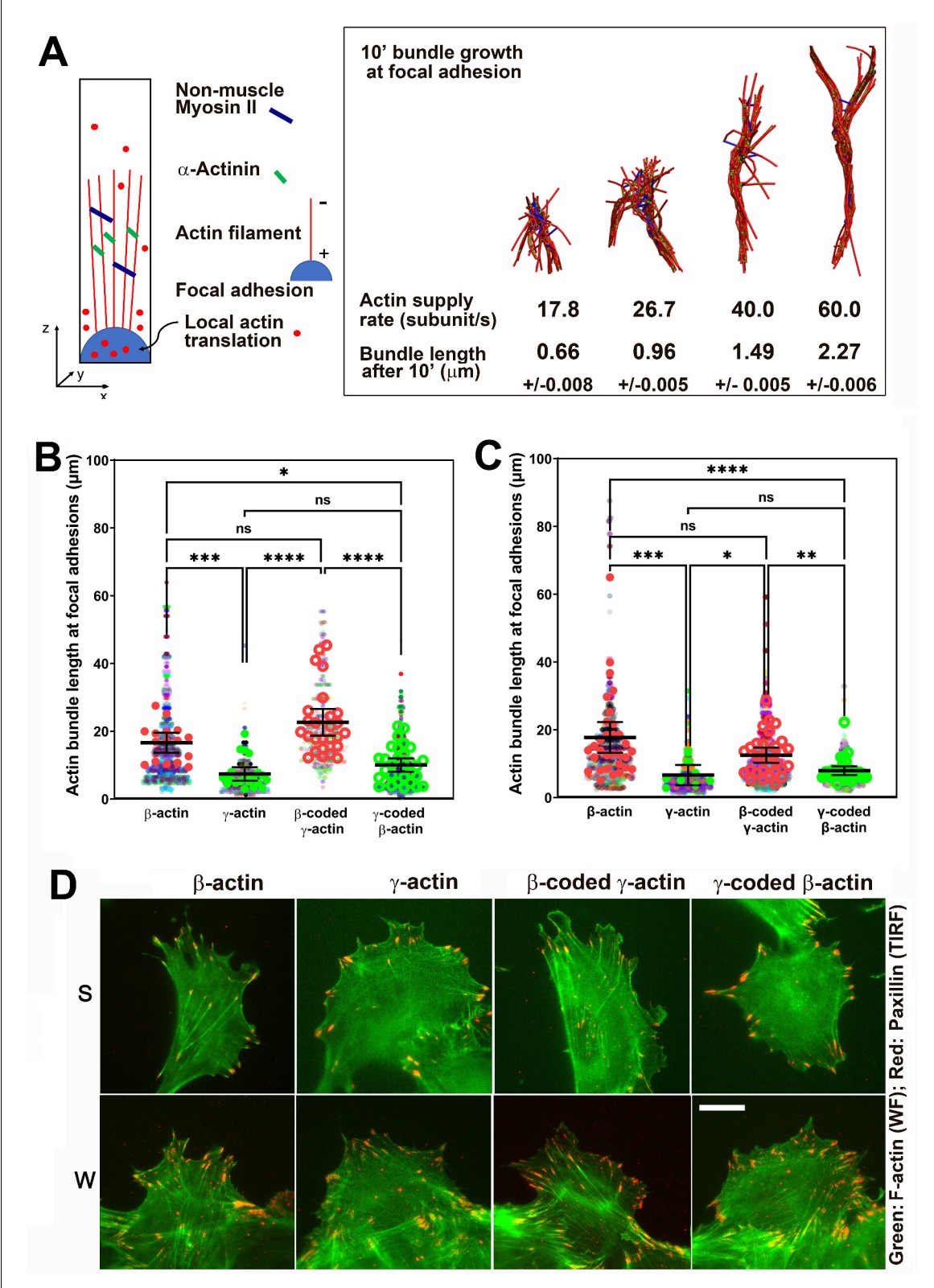

**Figure 5.** Faster β-actin subunit initial supply rate facilitates actin bundle formation at the focal adhesions to support normal cell migration. (**A**) Molecular simulations of actin filament growth at the focal adhesion at four different subunit supply rates, using the components listed on the right. Length of the actin bundle for each supply rate is indicated underneath, with SD listed for five independent simulations. All systems contain 0.012 μM non-muscle myosin II mini-filaments and 1.25 μM alpha-actinin crosslinkers. (**B, C**) Scatter plots showing the lengths of actin bundles emanating from

*Figure 5 continued on next page*

*Figure 5 continued*

focal adhesions in two separate experiments—single-cell cultures (B) and wound edge (C). Solid colors indicate individual cells (β-actin: 16, γ-actin: 21, β-coded γ-actin: 26, γ-coded β-actin: 30) or fields of view (β-actin: 31, γ-actin: 30, β-coded γ-actin: 8, γ-coded β-actin: 30), and transparent colors represent individual actin bundle length measurements. Error bars are mean ± 95% CI. Number of actin cables measured: (B) β-actin = 1715, γ-actin = 756, β-coded γ-actin = 482, and γ-coded β-actin = 2383; (C) β-actin = 1165, γ-actin = 1129, β-coded γ-actin = 1726, and γ-coded β-actin = 898. A non-parametric one-way analysis of variance (ANOVA) (Kruskal-Wallis) test for both single-cell cultures and wound edge gave a significant p-value (<0.0001). (D) Representative images of the single-cell cultures (S) and cells at wound edge (W) showing eGFP-actin (green) co-stained with focal adhesion marker paxillin (red). Scale bar, 20 μm.

The online version of this article includes the following figure supplement(s) for figure 5:

**Figure supplement 1.** Simulations of actin bundle growth from the focal adhesion.

prediction experimentally, we measured the length of eGFP-actin-decorated bundles emanating from paxillin-positive focal adhesion patches in cells stably expressing different eGFP-actin isoforms, using both the migrating cells at the edge of the wound and single cells (*Figure 5B–D*). In both types of cultures, actin bundles associated with the focal adhesion sites were markedly longer in β-actin-expressing cells, compared to those expressing γ-actin (*Figure 5B–D*). Moreover, these trends followed the actin coding sequence, rather than the amino acid sequence (*Figure 5B,C*). Thus, the slower subunit supply dictated by differences in translation elongation rates of β- and γ-actin coding sequences during the initial events of focal adhesion formation and maturation bears direct consequences to cell adhesion and migration.

## Discussion

Our study follows up on the recent discovery of the essential role of nucleotide, rather than amino acid, sequence in non-muscle actin isoform function and demonstrates for the first time that actin coding sequence, uncoupled from other gene elements, can directly affect cell behavior. We found that differences in β- and γ-actin coding sequences result in different ribosome elongation rates during their translation, leading to changes in cell spreading, focal adhesion anchoring, and cell migration speed. This study constitutes the first direct comparison of translation rates of two closely related proteins and the first demonstration that these translation rates can mediate their functions in vivo.

On the surface, it appears to be surprising that expression of the slower-translating γ-actin can make the cells move faster than the faster-translating β-actin. However, this result fits well into the context of the previously proposed localized bursts of β-actin translation, implicated in cell spreading and focal adhesion formation (*Condeelis and Singer, 2005*; *Katz et al., 2012*). Our data show that faster translating β-actin is required for generating stable focal adhesions, while the slower translating γ-actin leads to faster focal adhesion turnover. Importantly, our data also show that the effects mediated by slower translating γ-actin on cell migration manifest only when the *γ-actin* mRNA is targeted to the cell periphery via the β-actin zipcode sequence. Our study on codon profile-mediated actin isoform-specific translation differences, along with studies that showed actin isoform UTRs conferring isoform-specific mRNA localization with unique functional consequences (*Kislauskis et al., 1993*; *Moradi et al., 2017*), establish physiological roles to nucleotide elements in determining cellular phenotypes of isoactins.

We propose that slower translation at the leading edge makes γ-actin less capable of supporting and sustaining strong focal adhesions. Given that rapid polymerization of actin is required for lamellipodium protrusion and focal adhesion formation, one plausible role of localized fast actin translation at focal adhesions is to balance the competition between actin monomer pools required for protrusion and adhesion formation. In support, in γ-actin-expressing cells, most focal adhesions, while larger in area, do not appear to be visibly anchored by prominent actin bundles. In contrast, cells expressing β-actin contain long actin cables emanating from most of the focal adhesion sites. This difference impairs spreading without significantly changing protrusion dynamics. We propose that the faster migration rate is thus caused by weaker cell-substrate attachment in cells expressing γ-

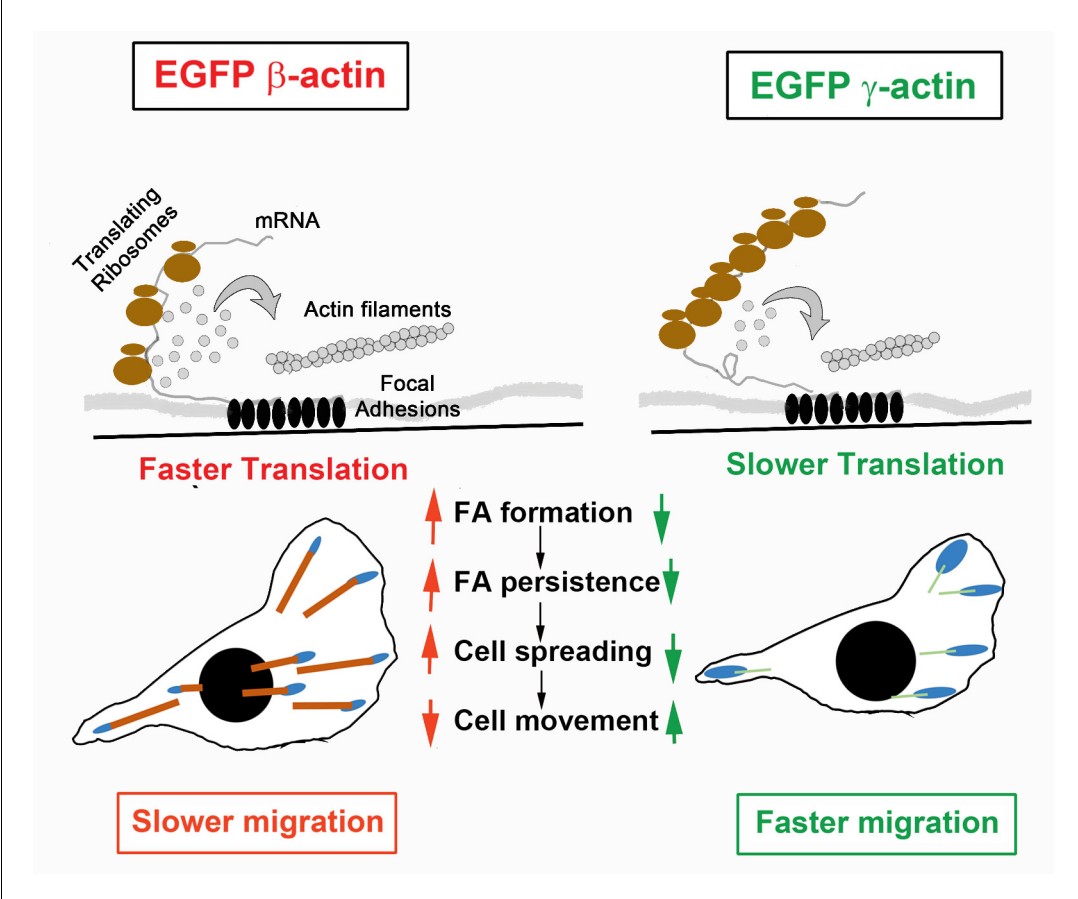

**Figure 6.** Hypothesis of cell migration regulation by actin isoforms' translation rates. Faster translating β-actin facilitates focal adhesion (FA) formation and persistence, resulting in increased cell spreading and a reduced rate of directional cell movement. Slower translating γ-actin has the opposite effect.

actin (*Figure 6*). Notably, this increase in cell speed is only evident in a wound-healing assay, where cells are collectively stimulated to migrate directionally, rather than randomly move around as typical for these cells in single-cell cultures of MEFs. It is likely that actin behavior in response to the strong signals for cells to polarize and move directionally during wound healing depends on local actin translation more critically than during random migration, where cells can change directions or remain stationary for extended periods of time and are not constrained in the direction of their polarization and motility. These constraints could potentially involve tension generated in dense cultures during collective migration (*Trepat et al., 2009*), as well as other forms of signaling in dense cultures.

While the local supply rate of actin subunits to the forming focal adhesion sites during cell migration is nearly impossible to measure experimentally with the currently available methods, the use of modeling and simulation enables us to vary this parameter and estimate the subunit supply rate that could make a difference in this process. It appears surprising that a twofold difference in translation elongation rate found in our study could exert such a pronounced effect on the length of the actin bundles forming locally at the focal adhesion sites, especially given the fact that polymerization-competent actin should exist everywhere in the lamellipodia. Our results suggest that the concentration of this polymerization-competent actin pool may be far lower than previously estimated, potentially due to the competition between different actin pools undergoing rapid polymerization, as well as the action of monomer sequestering proteins and/or posttranslational modifications that may prevent actin from incorporating into filaments (*Skruber et al., 2018*). It is also possible that, even with a higher actin concentration in the lamellipodia, focal adhesions compete with the monomer pool, thus requiring local translation of β-actin. Notably, these differences are expected to be even higher with the native non-eGFP-fused actin isoforms, which likely differ in their translation initiation rates in

addition to the difference in elongation rates we observed. These questions, and the exact interplay between newly synthesized and diffusible actin in cells, constitute an exciting direction of future studies.

Our previous work has shown that actin coding sequence leads to differential arginylation of β- and γ-actin (*Zhang et al., 2010*), and this arginylation is important for directional cell migration (*Karakozova et al., 2006*). In the present study, the use of N-terminal eGFP fusions likely excludes arginylation as a variable in our cell populations, since arginylation is believed to require an exposed β-actin N-terminus. The use of GFP-actin fusions also likely limits the types of effects we can observe, since these fusions are not fully functionally equivalent to the native actin and are impaired, for example, in formin nucleation (*Chen et al., 2012*). Thus, our constructs cannot substitute for all aspects of normal actin in cells, and this could explain the fact that the strongest effects we observe are related to cell adhesion and migration, the processes that are likely able to fully utilize eGFP-actin. Notably, in our cells, the endogenous β- and γ-actin are still present and still able to support cell migration, likely compensating for these types of functions, and potentially diminishing the observed phenotypes.

Thinking of the nucleotide sequence, rather than the amino acid sequence, as a determinant of actin function is a novel view that opens up many exciting questions. The present study demonstrates that coding sequence alone can play a significant role in cell behavior, but it does not exclude the possibility that other nucleotide-based elements of the actin gene also contribute to actin isoforms' global role in organism survival. For example, it has been shown that a unique intron in the γ-actin gene can contribute to its function (*Lloyd and Gunning, 1993*). Studies of the interplay of nucleotide- and amino acid-based determinants in actin functions constitute exciting future directions in the field.

# Materials and methods

## Key resources table

| Reagent type (species) or resource | Designation | Source or reference | Identifiers | Additional information |
|---|---|---|---|---|
| Strain, strain background (*Escherichia coli*) | NEB Stable (Stbl3) | New England Biosciences | C3040I | Chemically competent *E. coli* |
| Cell line (*Mus musculus*) | Spontaneously immortalized MEFs | This paper | | Mouse embryonic fibroblasts were derived from E12.5-E16.5 mouse embryos. The primary cells were cultured till spontaneously immortalized populations survived. |
| Transfected construct (synthetic) | pHR-scFv-GCN4-sfGFP-GB1-NLS-dWPRE | *Tanenbaum et al., 2014* | Addgene plasmid # 60906 | Plasmid containing Superfolder-GFP fused to single-chain variable fragment against GCN4 repeats used to localize nascent peptides from SINAPS constructs |
| Transfected construct (synthetic) | pBabe TIR1-9myc | *Holland et al., 2012* | Addgene plasmid # 47328 | Plasmid containing the ubiquitin ligase TIR, which targets auxin-induced degrons in nascent peptides from SINAPS constructs |

*Continued on next page*

*Continued*

| Reagent type (species) or resource | Designation | Source or reference | Identifiers | Additional information |
|---|---|---|---|---|
| Antibody | Anti-Paxillin (mouse monoclonal) | BD Transduction Laboratories | Cat#: 610619 | IF (1:100) |
| Recombinant DNA reagent | eTC GFP beta-actin full length (plasmid) | *Rodriguez et al., 2006* | Addgene plasmid # 27123 | Human *ACTB* promoter and 3'UTR containing plasmid with eGFP. This was used to construct the mouse β-, γ-, β-coded-γ-, and γ-coded-β-actin plasmids with β-actin 3'UTR. |
| Recombinant DNA reagent | eTC GFP beta-actin ΔZip (plasmid) | *Rodriguez et al., 2006* | Addgene plasmid # 27124 | Human *ACTB* promoter with no *ACTB* 3'UTR plasmid with eGFP. This was used to construct the mouse β-, γ-, β-coded-γ-, and γ-coded-β-actin plasmids without β-actin 3'UTR. |
| Recombinant DNA reagent | pUbC-FLAG-24xSuntagV4-oxEBFP-AID-baUTR1-24xMS2V5-Wpre (plasmid) | *Wu et al., 2016* | Addgene plasmid # 84561 | Plasmid containing Suntag, AID, β-actin 3'UTR, MS2 repeats. This was used to generate mouse SINAPS-β-actin and SINAPS-γ-actin reporters. |
| Recombinant DNA reagent | UbC NLS HA stdMCP stdHalo | *Voigt et al., 2017* | Addgene plasmid # 104999 | Plasmid containing MCP tandem dimer fused to HaloTag tandem dimer to visualize MS2 repeat containing RNA |
| Recombinant DNA reagent | pUbC-nls-ha-stdMCP-stdGFP | *Wu et al., 2016* | Addgene plasmid # 98916 | Plasmid containing MCP tandem dimer fused to GFP tandem dimer. Mouse vinculin sequence was cloned upstream of MCP to allow for tethering of MS2-containing RNA to focal adhesions. GFP was replaced by HaloTag. |
| Sequence-based reagent | Actb_F | This paper | PCR primers | GATCAAGATCAT TGCTCCTCCTG |
| Sequence-based reagent | Actb_F | This paper | PCR primers | AGGGTGTAAAAC GCAGCTCA |

*Continued on next page*

*Continued*

| Reagent type (species) or resource | Designation | Source or reference | Identifiers | Additional information |
|---|---|---|---|---|
| Sequence-based reagent | Actg1_F | This paper | PCR primers | GCGCAAGTACTC AGTCTGGAT |
| Sequence-based reagent | Actg1_R | This paper | PCR primers | TGCCAGGGCAA ATTGATACTTC |
| Sequence-based reagent | eGFP_F | This paper | PCR primers | GTGAAGTTCG AGGGCGACA |
| Sequence-based reagent | eGFP_R | This paper | PCR primers | TCGATGTTGT GGCGGATCTT |
| Sequence-based reagent | Tbp_F | This paper | PCR primers | TAATCCCAAG CGATTTGCTGC |
| Sequence-based reagent | Tbp_R | This paper | PCR primers | AGAACTTAGCT GGGAAGCCC |
| Sequence-based reagent | *eGFP* FISH probes | LGC Biosearch Technologies | VSMF 1015–5 | |
| Software, algorithm | FishQuant | *Mueller et al., 2013* | | Used for quantifying signals from SINAPS constructs in fixed cells |
| Software, algorithm | Airlocalize | *Lionnet et al., 2011* | | Used for quantifying signals from SINAPS constructs in FRAP experiments in live cells |

## Constructs

Constructs were generated using the eTC GFP beta-actin full-length plasmid (Addgene plasmid # 27123) and eTC GFP beta-actin ΔZip (Addgene plasmid # 27124), which were gifts from Robert Singer (*Rodriguez et al., 2006*). Plasmid #27123 encodes the human β-actin promoter, followed by TC-eGFP, a five-amino-acid (GSTSG) linker, and the full-length human β-actin complementary DNA (cDNA )containing β-actin 3′UTR (eTC GFP beta-actin full-length plasmid), followed by the transcription terminator bGH polyA, which carries the typical AAUAAA sequence responsible for transcription termination and polyadenylation of the mRNA. Plasmid #27214 is identical, except that it does not contain the beta-actin 3′UTR (eTC GFP beta-actin ΔZip). No 5′UTR or any other non-coding elements from the β-actin gene besides the promoter are included in these plasmids. To generate actin isoform-encoding constructs for this study, we used plasmid #27213 or #27214 as the backbone for the construct with and without β-actin 3′UTR, respectively. To obtain constructs expressing actin isoforms in this study, β-actin coding sequence (starting with the first ATG and ending with the terminator codon) was replaced with the corresponding sequence encoding mouse β-actin or mouse γ-actin (generating eGFP-β-actin and eGFP-γ-actin from plasmid #27213 and eGFP-β-actin Δ3′UTR and eGFP-γ-actin Δ3′UTR from plasmid #27214). For the codon-switched constructs, point mutations were introduced into the coding sequence of the actin isoforms as shown in *Figure 1* and described in *Vedula et al., 2017* to generate eGFP-β-coded γ-actin and eGFP-γ-coded β-actin. pUbC-FLAG-24xSuntagV4-oxEBFP-AID-baUTR1-24xMS2V5-Wpre was a gift from Robert Singer (Addgene plasmid # 84561) and was used to generate the actin isoform SINAPS reporters (*Wu et al., 2016*). The β- and γ-actin coding sequences were cloned in place of the oxEBFP sequence in the original construct. For fixed-cell imaging of ribosome load per mRNA, phage UbC NLS HA stdMCP stdHalo, a gift from Jeffrey Chao (Addgene plasmid # 104999) (*Voigt et al., 2017*), was used. For constructing the mRNA tether in live-cell real-time imaging of translation dynamics, pUbC-nls-ha-stdMCP-stdGFP, a gift from Robert Singer (Addgene plasmid # 98916) (*Wu et al., 2016*), was used to clone mouse vinculin sequence upstream of stdMCP, and stdGFP was replaced by HaloTag. pHR-scFv-GCN4-sfGFP-GB1-NLS-dWPRE, a gift from Ron Vale (Addgene plasmid # 60906) (*Tanenbaum et al., 2014*), was used for the NAP sensor. pBabe TIR1-9myc, a gift from Don Cleveland (Addgene plasmid # 47328) (*Holland et al., 2012*), was used for degrading the fully synthesized SINAPS construct.

## Generation of polyclonal stable cell populations

Spontaneously immortalized MEFs used in this project were obtained in the lab from E12.5-E16.5 mouse embryos and immortalized by continuous passaging in culture, using Dulbecco's modified Eagle's medium (DMEM) (Gibco) supplemented with 10% fetal bovine serum (FBS) (Gibco) as the tissue culture medium. These cells were produced and maintained in the lab and have not been independently authenticated, but they were continuously observed to maintain characteristic morphology of mouse embryonic fibroblasts. All mycoplasma tests conducted in the lab were negative.

To obtain the stable cell cultures described in this study, these cells were transfected with the linearized EGFP-actin constructs described above. Following G418 selection, GFP-positive cells were sorted using fluorescence activated cell sorting (FACS) and cultured. Live-cell imaging was carried out in FluorBrite DMEM (Life Technologies) culture media supplemented with 10% FBS (Sigma) and L-glutamine (Gibco).

HEK-293T cells were cultured in DMEM (Gibco) supplemented with 10% FBS (Gibco). Lipofectamine 2000 (Life Technologies) was used to transfect these cells with plasmids for generating either lentiviral particles, pMD.G, pPAX, and plasmid containing gene of interest, or retroviral particles, pCL10A and pBabe TIR1-9myc. Virus-containing medium was harvested and used to infect immortalized MEFs in the following order: first, either UbC-NLS-HA-stdMCP-stdHalo for fixed-cell imaging of number of NAPs/mRNA or Vinculin-stdMCP-Halo for live-cell dynamics of translation elongation, second, TIR1-9myc followed by puromycin selection of infected cells, third, scFv-GCN4-sfGFP-GB1-NLS, and lastly, either SINAPS-β-actin- or SINAPS-γ-actin-containing lentivirus. These polyclonal cells were used for imaging the number of NAPs on each of β- and γ-actin constructs. Indole-3-acetic acid was used at 500 µg/ml to induce degradation of fully synthesized SINAPS constructs. Janelia Fluor 646-tagged Halo ligand (Promega) was used at 200 nM final concentration to label *SINAPS-mRNA* in cells prior to fixation/imaging.

## Cell migration assays and imaging

Cell migration was stimulated by making an infinite scratch wound. The cells were allowed to recover for a period of 2 hr before imaging. Images were acquired using a X10 phase objective on a Lecia DMI 4000 equipped with a Hamamatsu ImagEM EMCCD camera. Images were captured every 5 min for 10 hr. Migration rates were measured as the area covered by the edge of the wound in the field of view per unit time using Fiji (NIH). For TIRF wound-healing experiments, cells were imaged on a Nikon Ti with a X100, 1.49 NA objective using the 488 nm laser and an Andor iXon Ultra 888 EMCCD.

## Fluorescence recovery after photobleaching

For all FRAP experiments, imaging was carried out on a Nikon Ti inverted microscope equipped with either an Andor iXon Ultra 888 EMCCD camera (0.13 µm/pixel—for imaging TIRF-FRAP and widefield whole-cell eGFP-actin FRAP) or an Andor iXon Ultra 897 EMCCD camera (0.16 µm/pixel—for imaging SINAPS-FRAP using a Yokogowa CSU X1 spinning disc confocal). Photobleaching was carried out with a Bruker miniscanner equipped with XY Galvo mirrors. The region of interest for photobleaching was defined using a freehand Region of Interest (ROI) manager in Nikon Instruments NIS elements software: an elliptical region encompassing an actin patch at the cell periphery for TIRF-FRAP, the entire cell for widefield whole-cell FRAP, and a single-pixel spot containing the translation site for SINAPS-FRAP. eGFP-actin-expressing cells were seeded on Matek glass bottom dishes and allowed to spread overnight. For photobleaching, the 488 nm laser was set to 80% power and used to bleach a defined eGFP-actin patch at the cell periphery with a dwell time of 400 µs/pixel. Images were acquired in the TIRF mode with the 488 nm laser set to 50% power and 200 ms exposure and an electron-multiplying (EM) gain of 200. Images were acquired at 3 s intervals for 12 s prebleach and 6 min post-bleach. The change in fluorescence intensity in a circle within an actin patch that was not bleached was used as a reference to account for photobleaching during acquisition. The change in intensity within a circle of the same area within the bleached actin patch was used to calculate the recovery curve. The obtained values were normalized to one at pre-bleach, and the resulting post-bleach curves were fit using non-linear regression to a single exponential fit in GraphPad PRISM.

For whole-cell eGFP-actin FRAP, the whole cell was outlined. For photobleaching, the 488 nm laser was set to 70% power with a dwell time of 70 µs/pixel. Acquisition was carried out using a 488 nm LED illumination from Spectra/Aura with 10% illumination intensity and 200 ms exposure with an EM gain of 300. Images were acquired every 10 s for a total of 10 min after bleaching. The recovery curves obtained were fit using a linear regression model in GraphPad PRISM.

For live-cell SINAPS-FRAP, cells expressing SINAPS-actin constructs were tethered using Vinculin-stdMCP-Halo (see sections 'Constructs' and 'Generation of polyclonal stable cell populations' above). For photobleaching, the 488 nm laser was set to 80% power with a dwell time of 1 ms/pixel. Images were acquired in the spinning disc confocal mode with the 488 nm laser set to 30% power and 35 ms exposure with an EM gain of 300. Images were acquired at 700 ms intervals for 30 s pre-bleach and 7 min post-bleach.

## Translation elongation rate measurements

FishQuant (*Mueller et al., 2013*) was used to detect NAP and mRNA signals. Spots in the NAP channel that were within 300 nm of a spot in the mRNA channel were considered bonafide NAPs and were used for estimating the integrated fluorescence intensity in both channels.

Airlocalize (*Lionnet et al., 2011*) was used to fit the signal from tethered NAPs. The integrated signal was recorded pre-bleach and post-bleach. These values were used to calculate the translation elongation rates of the two actin isoforms.

Assuming that beta-actin has the same elongation rate as Suntag, AID, and linkers, following the theoretical derivation of *Wu et al., 2016*, it is straight forward to show that the proportion of beta-actin contribution to recovery time and NAP/mRNA is $(L+(N + 1)/2)/(S + L + (N + 1)/2)$, in which N is the total number of Suntags, S is the beta-actin length in the unit of 1 Suntag, and L is the length of AID and linkers in the unit of 1 Suntag, shown as the gray bar in *Figure 4B and C*. It is not surprising to see from those figures that the ratios of recovery time to NAP/mRNA are similar for beta- and gamma-actin, since the initiation rates for both constructs should be the same, given the identical N-termini. Therefore, we calculated the variance-weighted geometric average of the two ratios, which is $T \sim 26.9$ s, and used it to combine the data from recovery time and NAP/mRNA to calculate the beta-actin elongation rate: $Rb=(S + L + (N + 1)/2)/t$ or $Rb=(S + L + (N + 1)/2)/n/T$, where t is recovery time and n is NAP/mRNA, for each data point, followed by geometric averaging. The contribution from Suntag, AID, and linkers to recovery time is $T0 = (L+(N + 1)/2)/Rb$, which is used to calculate the gamma-actin elongation rate: $Rg = S/(t-T0)$ or $Rg = S/(n*T-T0)$ for each data point, followed by geometric averaging. The results are shown in *Figure 4D*.

## Immunofluorescence staining and analysis

To quantify the amount of actin polymer, cells were seeded on coverslips in six-well plates at 20,000 cells/well overnight and fixed in 4% (w/v) paraformaldehyde (PFA) at room temperature for 30 min. Cells were then stained with phalloidin conjugated to AlexaFluor 594 (Molecular Probes). Images were acquired on Leica DM6000 at X40 and the total intensity of phalloidin per cell was measured using Fiji (NIH). To analyze focal adhesions in single cells, eGFP-actin-expressing cells were seeded on coverslips and allowed to adhere and spread overnight. Cells were then fixed in 4% (w/v) PFA at room temperature for 30 min followed by 0.5% Triton-X 100 treatment for 5 min. Cells were incubated with mouse anti-paxillin monoclonal antibody (BD Biosciences), followed by AlexaFluor 555-conjugated goat-anti-mouse secondary antibody (Life Technologies). Cells were imaged with Citifluor (Cytoskeleton Inc) anti-bleaching agent. To analyze cell spreading and cell area, Celltool was used to outline cell shapes and classify them and extract shape modes. The shape modes that captured 60% of the overall variability in the shape model were used to assess the distribution of cell shapes in a principle component analysis (PCA) plot. Additionally, a kernel density estimate of the marginal was used to plot the area of focal adhesions.

FISH *eGFP* mRNA probes (conjugated to Quasar 670 dye) were purchased from LGC Biosearch Technologies (VSMF 1015–5) and FISH was carried out as per manufacturers' protocol. Briefly, cells were seeded onto coverslips in six-well plates at 20,000 cells/well overnight and fixed in 4% (w/v) PFA at room temperature for 30 min followed by treatment with 70% alcohol at 4°C for 1 hr. Cells were incubated with 125 nM probes at 37°C overnight. Cells were stained with 4',6-diamidino-2-phenylindole (5 ng/ml) and mounted using Prolong Diamond (Life Technologies). Images were acquired

using Leica DM6000 at X40. Z-stacks were acquired, and blind deconvolution was carried out using Leica LAS X software.

## Real-time PCR

Cells were seeded onto 10-cm culture dishes and grown to confluence. RNA was isolated using RNeasy mini kit (Qiagen) and cDNA was synthesized using oligo dT primers with a first strand cDNA synthesis kit (Applied Biosystems). After standard curves were obtained, quantitative PCR (qPCR) was carried out using SybrGreen (Applied Biosystems) and the following primer sets. PCR was carried out on QuantStudio Flex 6 Real Time PCR system (Applied Biosystems). ΔΔCt method was used to estimate the relative expression levels of mRNA using *Tbp* as the reference transcript.

> Actb
> Forward primer: 5′ GATCAAGATCATTGCTCCTCCTG 3′
> Reverse primer: 5′ AGGGTGTAAAACGCAGCTCA 3′
> Actg1
> Forward primer: 5′ GCGCAAGTACTCAGTCTGGAT 3′
> Reverse primer: 5′ TGCCAGGGCAAATTGATACTTC 3′
> eGFP
> Forward primer: 5′ GTGAAGTTCGAGGGCGACA 3′
> Reverse primer: 5′ TCGATGTTGTGGCGGATCTT 3′
> Tbp
> Forward primer: 5′ TAATCCCAAGCGATTTGCTGC 3′
> Reverse primer: 5′ AGAACTTAGCTGGGAAGCCC 3′

## Simulations of actin filament growth at focal adhesions

Computational simulations of actin bundle growth from focal adhesions to predict the bundle length at different β- and γ-actin local supply rates were performed using a recently developed software MEDYAN (*Popov et al., 2016*). In brief, MEDYAN simulates actin networks by integrating the stochastic diffusion-reaction dynamics and mechanical relaxation of the cytoskeletal network. Diffusing molecular species, including actin monomers, unbound myosin motors, and unbound crosslinkers, are contained in a solution phase. Stochastic chemical reactions such as actin (de)polymerization and (un)binding of motors and linkers follow mass-action kinetics, and change the mechanical energy of the actin network. The net forces are then periodically relaxed using conjugate-gradient mechanical equilibration. This step also updates reaction rates of motor walking, motor unbinding, and linker unbinding, based on residue tension after minimization.

We used a $1 \times 1 \times 4$ μm$^3$ simulation box, containing non-muscle myosin II motors, alpha-actinin crosslinkers, actin monomers, and actin filaments growing from the bottom focal adhesion region. The focal adhesion region was presented as a hemisphere with 30 actin filaments attached. We tested and found that the actin filaments never grow longer than 4 μm in the z-direction, and thus, no length constraints on the actin bundles factored into the simulations. Actin filaments were only allowed to elongate at one end (the barbed end), while the elongation rate constant was averaged over filament polymerization rates and depolymerization rates of both the barbed end and the pointed end. The filament elongation was driven by the addition of actin monomers to the system, simulating the synthesis of actin monomers near the focal adhesion region. Multiple actin supply rates were tested at 50% increments based on the experimental measurements of the differences between β- and γ-actin synthesis rates. The simulations were run for 10 min to match the timescale of the experiments. The starting concentration of actin at the attachment site was assumed to be ~2 μM locally, creating an initial bundle at around 0.1 μm long. In the simulation, the majority (more than 90%) of actin for the filament growth was assumed to arise from the de novo subunit addition. The concentrations of myosin mini-filaments (0.012–0.021 μM) and alpha-actinin crosslinkers (1.25 μM) were chosen to ensure proper bundling of filaments (all model parameters are listed in *Table 1*).

To determine the actin bundle length, we measured the F-actin distribution along the Z-axis and defined the actin bundle length as the width of central 80% of the F-actin distribution (*Figure 5—figure supplement 1A*). Although the length measured in simulations was much shorter than

**Table 1.** Model parameters.

| Parameter name | Value | Reference and notes |
|---|---|---|
| Filaments | | |
| Filament polymerization and depolymerization rates | $\kappa_{on}$ = 12.9 (11.6 + 1.3) $\mu M^{-1}s^{-1}$ <br> $\kappa_{off}$ = 2.2 (1.4 + 0.8) $s^{-1}$ | (*Fujiwara et al., 2007*) Taking both barbed ends and pointed ends into account |
| Filament bending constant | 672.5 pN·nm | (*Ott et al., 1993*) Bending constant between connecting cylinders[*] |
| Filament stretching constant | 100.0 pN/nm | (*Popov et al., 2016*) Stretching constant of cylinder |
| Motors | | |
| Binding rate constant | 0.2 $s^{-1}$ per head | *Stam et al., 2015* |
| Duty ratio | 0.1 | *Stam et al., 2015* |
| Number of motor heads per mini-filament | 15–30 | *Verkhovsky and Borisy, 1993* |
| Stretching constant | 56.0 pN/nm | *Popov et al., 2016* |
| Characteristic stall force | 15.0 pN per head | *Popov et al., 2016* |
| Characteristic motor unbinding force | 12.6 pN per head | *Erdmann et al., 2013* |
| Crosslinkers | | |
| Binding rate constant | 0.7 $\mu M^{-1}s^{-1}$ | *Wachsstock et al., 1993* |
| Unbinding rate constant | 0.3 $s^{-1}$ | *Wachsstock et al., 1993* |
| Stretching constant | 8.0 pN/nm | *DiDonna and Levine, 2007* |
| Characteristic linker unbinding force | 17.2 pN | *Ferrer et al., 2008* |

[*] Filaments are discretized into cylinders, and the length of cylinders ranges from 2.7 nm (1 subunit) to a maximum of 108 nm (40 subunits). Bending is only allowed between two connecting cylinders.

that in the experiments, the beta-actin bundles were ~50–80% longer than gamma-actin bundles, in agreement with the experimental measurements.

## Acknowledgements

We thank members of the Kashina lab for helpful and stimulating discussions. This work was supported by NIH grant R35GM122505 to AK and R01CA201340 and R01EY028450 to YJ. Simulations of actin bundle growth at the focal adhesion are a result of the collaboration between Kashina and Jiang labs initiated at the NSF-sponsored workshop funded by award no. MCB-1411898. Simulations were supported by NSF grants CHE-1800418 and PHY-1806903 to GP and were performed on Deepthought2 Supercomputer at the University of Maryland.

## Additional information

### Funding

| Funder | Grant reference number | Author |
|---|---|---|
| National Institutes of Health | R35GM122505 | Anna Kashina |
| National Institutes of Health | R01CA201340 | Yi Jiang |
| National Institutes of Health | R01EY028450 | Yi Jiang |
| National Science Foundation | CHE-1800418 | Garegin Papoian |
| National Science Foundation | PHY-1806903 | Garegin Papoian |

The funders had no role in study design, data collection and interpretation, or the decision to submit the work for publication.

## Author contributions
Pavan Vedula, Conceptualization, Data curation, Formal analysis, Validation, Investigation, Visualization, Methodology, Writing - original draft, Writing - review and editing; Satoshi Kurosaka, Conceptualization, Resources, Investigation, Visualization, Methodology; Brittany MacTaggart, Investigation, Visualization; Qin Ni, Formal analysis, Validation, Visualization, Methodology; Garegin Papoian, Resources, Software, Formal analysis, Supervision, Funding acquisition; Yi Jiang, Resources, Software, Formal analysis, Supervision, Funding acquisition, Investigation, Methodology; Dawei W Dong, Conceptualization, Data curation, Formal analysis, Validation, Investigation, Methodology; Anna Kashina, Conceptualization, Resources, Data curation, Formal analysis, Supervision, Funding acquisition, Validation, Investigation, Visualization, Methodology, Writing - original draft, Project administration, Writing - review and editing

## Author ORCIDs
Pavan Vedula ⓘD https://orcid.org/0000-0002-9914-0008
Satoshi Kurosaka ⓘD http://orcid.org/0000-0002-4365-9003
Brittany MacTaggart ⓘD http://orcid.org/0000-0001-7674-6042
Anna Kashina ⓘD https://orcid.org/0000-0002-0243-6866

## Decision letter and Author response
Decision letter https://doi.org/10.7554/eLife.68712.sa1
Author response https://doi.org/10.7554/eLife.68712.sa2

# Additional files

## Supplementary files
• Transparent reporting form

## Data availability
Data generated or analyzed during this study are included in the manuscript and supporting files. Raw images and videos for the main text figures are available at the Dryad depository with the following unique identifier: https://doi.org/10.5061/dryad.z34tmpgd2.

The following dataset was generated:

| Author(s) | Year | Dataset title | Dataset URL | Database and Identifier |
|---|---|---|---|---|
| Kashina A, Vedula P | 2021 | Data from: Different translation dynamics of $\beta$- and $\gamma$-actin regulates cell migration | http://dx.doi.org/10.5061/dryad.z34tmpgd2 | Dryad Digital Repository, 10.5061/dryad.z34tmpgd2 |

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
