## [Decision Letter]

**Acceptance summary:**

This work provides good evidence that the mRNAs for the actin isoforms mainly differ in their spatial expression of protein, rather than the protein itself and this has profound effects on the cell physiology.

**Decision letter after peer review:**

Thank you for submitting your article "Different translation dynamics of β-and γ-actin regulates cell migration" for consideration by *eLife*. Your article has been reviewed by 2 peer reviewers, and the evaluation has been overseen by a Reviewing Editor and Anna Akhmanova as the Senior Editor. The following individuals involved in review of your submission have agreed to reveal their identity: R Dyche Mullins (Reviewer #2); Peter W Gunning (Reviewer #3).

Essential revisions:

1. The details of the gene construct used to build the vectors is not clear in Methods. How much 5' upstream sequence of the β-actin gene is used and from what species (chicken, mouse, human)? Are any introns present? Is the full 5'UTR of β-actin included? Is there a linker between GFP and the amino terminus of actin? What provides the transcription termination down stream of the polyadenylation site? This is all important information to enable reproduction of the study.

2. The authors refer to cell lines throughout the paper. Transfected cell lines relate to a clonal origin whereas these are polyclonal populations of cells. I recommend avoidance of the term cell lines to avoid confusion.

3. In Figure 1 the legend refers to constructs that lack the β-actin 3'UTR targeting sequence, but this is not shown in the figure. Is only the targeting sequence eliminated or is the entire β-actin 3'UTR removed? If it is the latter, what is used to provide a 3'UTR? Results are mentioned in the text that demonstrate that the functional differences are only observed with the targeting sequence intact in the constructs. This is an important point and addressed in Figure S5, however, it is not clear if removal of the targeting sequence from the γ-actin gene construct produces a significant decrease in motility. Is the difference statistically significant?

4. The images in Figure 2 would benefit from quantitation of focal adhesion lifetimes and morphology (beyond area convincingly covered in Figure 3) to support the conclusions that both are impacted differently by the nucleotide sequences of β- and γ-actin.

5. The images in Figure 3 suggest that unlike β-actin with both peripheral and internal focal adhesions, the γ-actin nucleotide sequence promotes almost exclusively only peripheral focal adhesions. Is this widespread in the cells or does it reflect atypical images in the figure?

6. In the Discussion the authors argue that local synthesis is important for focal adhesion assembly and that there is low availability of actin for polymerisation from the lamellipodium. This seems unlikely because of the rapid polymerisation of actin in the lamellipodium. Could this rather reflect competition between the lamellipodium and focal adhesions for polymerisation competent actin?

7. The results of Kislauskis and Singer (1997, J Cell Biol) and their work on the spatial distribution of β and α actin isoforms (J. Cell Bio, 1993) and the effect on cell phenotype (J. Cell Bio 1994) should be incorporated into the Discussion since they have addressed the significance of peripheral targeting of β-actin.

8. More information on statistics would improve the presentation. For example in the scratch assays in Figure 1 the N values are reported as 20 and 22 (for β- and γ-actin respectively) but it is unclear whether this reflects the number of scratches, the number of coverslips, or (unlikely) the number of days on which the experiment was repeated. All of these numbers would be useful for judging the significance of the result. Also in the box-and-whisker plots the authors should indicate the replicate-to-replicate (day-to-day) variation in the results.

9. The above comment applies generally to the presentation of quantitative cell biological data in the paper. In Figure 3, for example, N is the number of focal adhesions but it is unclear how many cells, coverslips, and/or biological replicates are included in the data. The bar graphs should be replaced by graphics that illustrate the variation between biological replicates (one way to do this would be 'super-plots' https://pubmed.ncbi.nlm.nih.gov/32346721/ – note that other methods are available). Ditto for quantitative data in Figures 4 and 5 and all the Supplementary Figures.

---

## [Author Response]

Essential revisions:1. The details of the gene construct used to build the vectors is not clear in Methods. How much 5' upstream sequence of the β-actin gene is used and from what species (chicken, mouse, human)? Are any introns present? Is the full 5'UTR of β-actin included? Is there a linker between GFP and the amino terminus of actin? What provides the transcription termination down stream of the polyadenylation site? This is all important information to enable reproduction of the study.

We agree that this section of the Methods was not written clearly enough. In response to this comment, we updated it with the requested details, including the description of the original plasmids, and the regions replaced in these plasmids to generate the constructs for the study. In brief, the only 5′ upstream non-coding sequence used was the human β-actin promoter. No introns were present. There was a short 5-aminoacid linker between GFP and the N-terminus of actin. Transcription terminator used was bGH polyA, which carries the typical AAUAAA sequence responsible for transcription termination and polyadenylation of the mRNA. All these details of the original plasmids and the modifications we introduced were expanded in the first section of the Materials and methods.

2. The authors refer to cell lines throughout the paper. Transfected cell lines relate to a clonal origin whereas these are polyclonal populations of cells. I recommend avoidance of the term cell lines to avoid confusion.

This is a good point. All mentions of “cell lines” in the manuscript have been edited to either “cell cultures”, “cell populations” or “stable polyclonal cell populations” to reflect their non-clonal origin.

3. In Figure 1 the legend refers to constructs that lack the β-actin 3'UTR targeting sequence, but this is not shown in the figure. Is only the targeting sequence eliminated or is the entire β-actin 3'UTR removed? If it is the latter, what is used to provide a 3'UTR? Results are mentioned in the text that demonstrate that the functional differences are only observed with the targeting sequence intact in the constructs. This is an important point and addressed in Figure S5, however, it is not clear if removal of the targeting sequence from the γ-actin gene construct produces a significant decrease in motility. Is the difference statistically significant?

We have eliminated the mention of constructs lacking 3′UTR from the Figure 1 legend.

We used different parent plasmids to generate constructs with and without 3’ UTR. In the plasmids with 3′UTR, the entire 3′UTR from β actin was present. In the plasmids lacking 3′UTR the entire sequence was absent. In both cases, there was the bGH ployA on the 3′ end of the construct, and in the δ 3′UTR plasmid this sequence was the only 3′ UTR element. These details have been added to the Methods section. Of note, this strategy is routine in the actin mRNA targeting studies and these plasmids have been well characterized by R. Singer’s group.

We modified the supplementary figure S5 to show a comparison of migration rates in cells expressing γ-actin with and without β-actin 3′UTR and included the information on the statistical significance, as well as individual repeats.

4. The images in Figure 2 would benefit from quantitation of focal adhesion lifetimes and morphology (beyond area convincingly covered in Figure 3) to support the conclusions that both are impacted differently by the nucleotide sequences of β- and γ-actin.

Focal adhesion lifetimes were quantified, and the quantification included in Figure 2. Focal adhesion morphology described by the aspect ratio was estimated and results plotted in supplementary Figure S13C. While visually larger focal adhesions appear less elongated in γ and γ-coded-β actin cells, the aspect ratio quantification does not show any trend with either the coding sequence or the amino acid sequence.

5. The images in Figure 3 suggest that unlike β-actin with both peripheral and internal focal adhesions, the γ-actin nucleotide sequence promotes almost exclusively only peripheral focal adhesions. Is this widespread in the cells or does it reflect atypical images in the figure?

To address this point, we quantified the total number of focal adhesions per cell in each transfected culture, as well as the number of focal adhesions away from the leading edge in each culture. Both parameters show a change consistent with the actin isoforms’ coding sequence, and the differences in focal adhesion numbers are highly statistically significant. These data are now presented in Figure S13A and Figure S13B.

6. In the Discussion the authors argue that local synthesis is important for focal adhesion assembly and that there is low availability of actin for polymerisation from the lamellipodium. This seems unlikely because of the rapid polymerisation of actin in the lamellipodium. Could this rather reflect competition between the lamellipodium and focal adhesions for polymerisation competent actin?

This is a very good possibility. The question of availability of monomeric actin in the cell and the various mechanisms that could potentially limit this availability has been a subject of a number of recent studies, and with all the work about how various actin networks are in competition for monomers, this explanation appears likely. We have included this possibility into the discussion.

7. The results of Kislauskis and Singer (1997, J Cell Biol) and their work on the spatial distribution of β and α actin isoforms (J. Cell Bio, 1993) and the effect on cell phenotype (J. Cell Bio 1994) should be incorporated into the Discussion since they have addressed the significance of peripheral targeting of β-actin.

Agreed. We have expanded the section on 3′UTR in the discussion and included the references mentioned.

8. More information on statistics would improve the presentation. For example in the scratch assays in Figure 1 the N values are reported as 20 and 22 (for β- and γ-actin respectively) but it is unclear whether this reflects the number of scratches, the number of coverslips, or (unlikely) the number of days on which the experiment was repeated. All of these numbers would be useful for judging the significance of the result. Also in the box-and-whisker plots the authors should indicate the replicate-to-replicate (day-to-day) variation in the results.

This is a great point. We modified most of the charts in the manuscript to include data on biological replicates and technical replicates, which are now marked by differently styled dots in the plots.

9. The above comment applies generally to the presentation of quantitative cell biological data in the paper. In Figure 3, for example, N is the number of focal adhesions but it is unclear how many cells, coverslips, and/or biological replicates are included in the data. The bar graphs should be replaced by graphics that illustrate the variation between biological replicates (one way to do this would be 'super-plots' https://pubmed.ncbi.nlm.nih.gov/32346721/ – note that other methods are available). Ditto for quantitative data in Figures 4 and 5 and all the Supplementary Figures.

We agree. Following this suggestion, we used super-plots where possible to show variation among cells with the populations.